# Enhancing Model Interpretability with Local Attribution over Global Exploration

## ABSTRACT

In the field of artificial intelligence, AI models are frequently described as 'black boxes' due to the obscurity of their internal mechanisms. It has ignited research interest on model interpretability, especially in attribution methods that offers precise explanations of model decisions. Current attribution algorithms typically evaluate the importance of each parameter by exploring the sample space. A large number of intermediate states are introduced during the exploration process, which may reach the model's Out-of-Distribution (OOD) space. Such intermediate states will impact the attribution results, making it challenging to grasp the relative importance of features. In this paper, we firstly define the local space and its relevant properties, and we propose the Local Attribution (LA) algorithm that leverages these properties. The LA algorithm comprises both targeted and untargeted exploration phases, which are designed to effectively generate intermediate states for attribution that thoroughly encompass the local space. Compared to the state-of-the-art attribution methods, our approach achieves an average improvement of 38.21% in attribution effectiveness. Extensive ablation studies within our experiments also validate the significance of each component in our algorithm. Our code is available at: https://anonymous.4open.science/r/LA-2024

## CCS CONCEPTS

• **Security and privacy** → **Trusted computing**.

## KEYWORDS

XAI, Interpretability, Attribution

## 1 INTRODUCTION

Recent years have witnessed the emergence of deep learning, which has significantly advanced the development of artificial intelligence (AI). It has enabled computers to learn from extensive data and achieve remarkable performance in areas such as image recognition and natural language processing [9, 11], contributing to almost every aspect of our daily life. For instance, in healthcare, deep learning aids doctors in diseases diagnosis and treatments planning [4]. In transportation, it powers autonomous vehicles that navigate cities and highways safely and efficiently [16]. It also enhance customer service by answering inquiries and solving issues [1]. Moreover, AI

*ACM MM, 2024, Melbourne, Australia*

© 2024 Copyright held by the owner/author(s). Publication rights licensed to ACM.
ACM ISBN 978-x-xxxx-xxxx-x/YY/MM
https://doi.org/10.1145/nnnnnnn.nnnnnnn

is becoming pivotal in industries such as finance and manufacturing, where it optimizes operations and boosts efficiency [3, 13, 29].

Despite AI's increasing practice, its models are often regarded as 'black box', reflecting the transparency and trust concerns in understanding how these models make decisions. It leads to several significant challenges and potential issues. First, it undermines users' trust in AI systems. In critical sectors like healthcare and finance, a transparent decision-making process is essential for users to trust the recommendations [12, 14]. A lack of trust will greatly reduce the practical value of even the state-of-the-art technology [33]. Secondly, it complicates the identification and mitigation of errors or biases. For example, addressing gender or racial bias in AI assisted hiring is challenging without insights into the decision-making criteria [28]. Furthermore, the 'black box' nature of AI poses challenges for legal and ethical responsibility [6]. In cases where AI systems cause harm or disputes, pinpointing responsibility is difficult if the principles behind the behavior cannot be explained. Lastly, this opacity can also hinder the regulation and public oversight of AI technologies, leading to technological developments that deviate from societal ethics and values.

To address these challenges, Explainable AI (XAI) becomes a trending topic for research, aiming to increase the transparency and interpretability of artificial intelligence decision-making processes. LIME is one of the earliest methods which approximates the behavior of complex models around given inputs [19]. However, it fails to provide comprehensive and precise insights, and can sometimes produce misleading explanations due to the reliance on simplified assumptions about model behavior. Later approaches, such as Grad-cam [20] and Score-cam [30], which use gradients information, are limited by model structure and do not produce fine-grained (input-dimension consistent) results. Introduced in Integrated Gradients (IG) [25] with the axiomatic properties, *Sensitivity and Implementation Invariance*, attribution method marks a significant advancement in XAI. As a more robust approach, it provides high-resolution, fine-grained explainability and are not limited to model structure, allowing for precise attribution of model results based on rigorous axiomatic principles.

Current attribution methods generally calculate the importance of each dimension by accumulating gradients over intermediate states [17, 25, 31, 35], which ensures compliance with axioms of Sensitivity and Implementation invariance [25]. However, they often fail to address the plausibility of these intermediate states. Considering the extensive input space neural networks encounter, it is impractical to accurately assess every potential state. In this work, we firstly investigate this phenomenon and define a space that neural networks are responsible for as the In-Distribution (ID) space, whereas the space they are not responsible for is termed Out-Of-Distribution (OOD) space. We observe that, most intermediate states utilized in current attribution algorithms often fall within the defined OOD space, which has led to attribution errors.

To further investigate what types of intermediate states will contribute to attribution results, we discuss two research questions:

- **RQ1**: Does the significance of the attribution results still hold if there is a critical deviation in the features?
- **RQ2**: Can we still accurately assess the importance of remaining features when key features are excluded?

In addressing these research questions, we introduce the concept of local spaces, where attribution approves to be valuable and precise. Inspired by MFABA [35] and AGI [17], we explore a combination of targeted and untargeted adversarial attacks in local spaces. We provide a thorough analysis from an optimization perspective on how these attacks can be integrated and their contribution to attribution exploration. Building on this analysis, we propose the Local Attribution (LA) method, for which we provide the detailed mathematical derivations and proofs demonstrating its compliance with attribution axioms. Our contributions are outlined as follows:

- We identify that current attribution methods often assign intermediate states to spaces not contributing to attribution results. To address this issue, we introduce two research questions and the concept of attribution local spaces.
- We design a Local Attribution (LA) method to ensure that each intermediate state remains within the attribution local space, and we provide detailed mathematical derivations and proofs of its axiomatic properties.
- With extensive experiments, we demonstrate the effectiveness of the LA method. Compared to other state-of-the-art methods, LA algorithm improves the Insertion Score by an average of 38.21% and reduces the Deletion Score by 11.52%, significantly outperforming existing technologies.
- We have released the implementation code of the LA algorithm, to facilitate the exploration, validation and improvement together with other XAI researchers.

## 2 RELATED WORK

In this section, we explore different methods used for explaining Deep Neural Networks (DNNs) and provide a critical discussion of these approaches, which are grouped by three types: local approximation methods, gradient-based attribution methods, and adversarial-sample-based attribution methods.

### 2.1 Local Approximation Methods

Local approximation methods seek to understand the behavior of the original model near specific inputs by constructing an approximate, more interpretable model. A well-known method is LIME [19], which approximates local explainability by using multiple interpretable structures near the sample. However, LIME's local explainability requires assumptions, which may not always be accurate. Moreover, LIME can be time-consuming for individual samples. While rudimentary for neural network applications, LIME has been foundational in advancing local explainability methods. Following developments include Layer-wise Relevance Propagation [2] and DeepLIFT [21]. DeepLIFT quantifies the importance of features by comparing the differences between input features and predefined reference points. Although DeepLIFT performs well in local explanation of nonlinear models, its high sensitivity to the choice of reference points can lead to inconsistencies in attribution

results. Additionally, DeepLIFT does not satisfy the Implementation invariance axiom proposed in IG [25], leading to potential biases.

### 2.2 Gradient-based Attribution Methods

Training neural networks inherently utilize gradients, which has inspired the gradient-based methods that use model gradient information to explain decisions. Early methods like Saliency Map (SM) [22] identify the most important features for model predictions by calculating the gradients of input features relative to the model output. However, SM is prone to gradient saturation, resulting in unstable attribution results, and it does not meet the Sensitivity axiom mentioned in subsequent IG [25], meaning it can yield a zero attribution even if the model output changes. Later methods such as Grad-cam [20] and Score-cam [30] use intermediate layer gradient information but cannot provide high-resolution fine-grained explainability results, and thus cannot be considered true attribution methods (refer to Section 3.1 problem definition).

The IG method addresses the insufficient gradient issue of SM by integrating gradients along the path from baseline to input, introducing the axioms of sensitivity and implementation invariance, which are fundamental guarantees for attribution algorithms. Our design also provides proofs of compliance with these axioms. However, IG's main challenge lies in its high computational cost, requiring multiple forward and backward passes. To improve computational efficiency, Fast IG (FIG) [8] optimizes the IG method by improving numerical integration techniques to speed up the attribution process. Although this optimization enhances efficiency, the approximate nature of numerical integration might introduce new errors, affecting the accuracy of attribution results. Additionally, Expected Gradients (EG) [5] provides a more stable and consistent assessment of feature importance by considering multiple baselines and averaging their gradients, improving the IG method. However, a limitation of the EG method is its assumption that contributions from different baselines are equal, which may not be suitable for all types of data and model structures, thus affecting the generalizability of its explanations. SmoothGrad (SG) [24] improves the smoothness and stability of attribution results by adding random noise to inputs, reducing the noise in single gradient calculations. Despite these improvements, the addition of noise may also mask understanding of subtle features important to the model's decision-making process, thus reducing the accuracy of explanations. Guided IG (GIG) [10] combines the principles of IG and guided backpropagation by selectively backpropagating gradients to enhance interpretability. However, GIG's limitation is that it may overemphasize features directly related to specific categories while ignoring indirect features that are equally important to model decisions, somewhat limiting its ability to provide comprehensive explanations. This school of attribution algorithms based on IG is limited by the choice of baseline in the attribution path, introducing a significant amount of irrelevant noise.

### 2.3 Adversarial-sample-based Attribution Methods

Adversarial-sample-based attribution methods provide deep explanations of models by generating adversarial samples and exploring model decision boundaries, meaning the attribution process no

longer relies on manually specified baseline points. Adversarial Gradient Integration (AGI) [17] is a representative work that uses adversarial samples to explore decision boundaries and improves attribution performance through nonlinear path integral gradients. While AGI offers an innovative method of explanation, its performance highly depends on the quality of the adversarial samples, which may not be stable in some cases.

Boundary-based Integrated Gradients (BIG) [31] introduces a boundary search mechanism to optimize the baseline selection, thereby obtaining more accurate feature attributions. However, BIG relies on a linear integration path, which may limit its ability to capture the nonlinearity and complexity in model decision. AttEXplore [34] improves feature attribution by combining adversarial attacks with model parameter exploration, emphasizing the ability to transition between different decision boundaries. Although AttEXplore shows foresight in enhancing the generalization ability of model explanations, its high computational complexity may limit its application on large-scale models and datasets. MFABA (More Faithful and Accelerated Boundary-based Attribution) [35] enhances the accuracy and computational efficiency of explanations through second-order Taylor expansion and decision boundary exploration, particularly suited for complex model explanations. Nonetheless, its reliance on higher-order derivatives may increase the computational burden, especially when dealing with large deep learning models. This class of adversarial-sample-based attribution methods introduces a large number of intermediate states from the OOD space during the adversarial attack process, as shown in Figure. 1, affecting the accuracy of attribution (discussed in Section 3.2).

## 3 METHOD

In this section, we define the attribution task, the local properties of attribution, and the algorithmic procedure of the LA (Local Attribution) method. Ensuring local properties is key to the rationality of attribution results, and within these constraints, it is still possible to achieve results that satisfy the remaining axioms of attribution. We will describe these in detail below and provide rigorous mathematical derivations. Additionally, the LA algorithm consists of two parts: targeted and untargeted attribution, which can be combined under the premise of maintaining local properties.

### 3.1 Problem Definition

Given neural network parameters $w \in \mathbb{R}^n$ and a sample $x \in \mathbb{R}^n$ to be attributed, we aim to use an attribution method to obtain attribution results $A(x) \in \mathbb{R}^n$, where $A_i(x)$ represents the importance of the $i$-th feature dimension. The greater the attribution result, the more important the dimension is for the model's decision. We use $f(x) \in \mathbb{R}^c$ to represent the model output, where $c$ denotes the number of classes.

### 3.2 Local Space of Attribution

Before introducing local properties, we present a critical research question: **RQ1: After significant deviation of features, does the importance assessment of altered features have referential significance?** Current mainstream attribution algorithms overlook this question. To illustrate, consider a toy example where a data sample $x$ has four dimensions $x = [6, 8, 6, 10]$. During the use

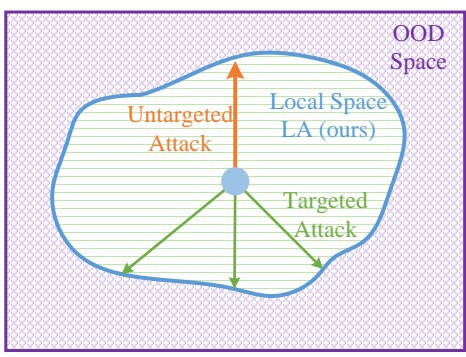

Figure 1: A vast amount of Out-of-Distribution (OOD) space exists outside the defined Local Space, where samples within the OOD space lack guidance for attribution. Furthermore, the use of both untargeted and targeted attacks enables the exploration of a possibly comprehensive Local Space. This aspect was discussed in depth from the perspective of the loss function in Section 3.3.

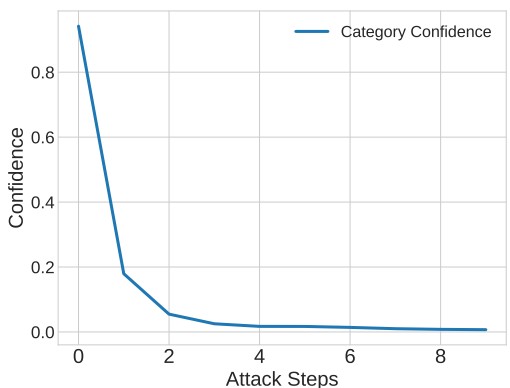

Figure 2: When adversarial attacks exceed two iterations, the model essentially lacks the current category's characteristics, and subsequent samples in the OOD space no longer guide the attribution algorithm meaningfully.

of IG [25], gradients of intermediate variables accumulated from the sample to the baseline are considered. Suppose there is only one intermediate state, and the baseline is $b = [0, 0, 0, 0]$. Thus, the intermediate variable $x'$ lies between $x$ and $b$ at $x' = [3, 4, 3, 5]$. At this moment, we need to compute the gradient information, but is this gradient information truly valuable? Given the vast input space neural networks face—a space so large it's impossible to traverse fully—it means that models generally need only be responsible for In-Distribution (ID) samples, and most of the space filled with Out-Of-Distribution (OOD) samples is meaningless. Similarly, in the OOD space, gradients will lack instructive significance because it is virtually impossible for the model to encounter $x' = [3, 4, 3, 5]$ in tasks, and $x'$ at this point cannot sustain the model's decision, placing $x'$ within the OOD space. This means that assessing sample

feature importance in scenarios where features undergo significant deviations and cannot maintain model decisions introduces too much extraneous information from spaces the model is not responsible for. Other methods like MFABA [35] and AGI [17] use adversarial attacks to obtain intermediate states $x'$, but when adversarial attacks are sufficiently frequent, $x'$ will ultimately fall into OOD space. As shown in Figure. 2, after two or more adversarial attacks, the attack samples are insufficient to maintain the model's decisions, leading to serious attribution biases.

We need further definition on what samples are considered to be in the ID space. In Multiplicative Smoothing (MuS) [32], the explainable method stable can be seen as when enough important features are satisfied, adding additional features will not affect the original model's decision. That is, the model has obtained enough important features to maintain the current decision, meaning these crucial features are key to keeping the target within the ID space. This also leads to **RQ2: When important features are disregarded, can the remaining features still be correctly assessed for importance?**

As shown in Figure. 3, since assessing feature importance is meaningless without important features, we must ensure that these features do not change during the assessment. Unfortunately, we cannot determine which features are important during the assessment phase (importance can only be confirmed after, not during the assessment, as they influence each other). Here, we provide the definition of when an intermediate state is considered to be in the Out-of-Distribution (OOD) space: **An intermediate state is in the OOD space if it cannot maintain the same model decision as the original state.** The only remaining option is to ensure that all features are assessed within a **local space where no significant deviations occur**.

Next we give the definition of attribution local space:

**THEOREM 3.1 (LOCAL SPACE).** *Given a sample $x$, the $\epsilon$-Local Space of $x$, denoted as $B_\epsilon(x)$, is defined as:*

$$B_\epsilon(x) = \{\tilde{x} \mid |\tilde{x}_i - x_i| \le \epsilon_i\} \tag{1}$$

*where $\epsilon \in \mathbb{R}^n$ and $\epsilon_i = \frac{x_i}{s}$, with $s$ being a hyperparameter that controls the size of the local space (Spatial Range).*

We assume that the importance assessment within the local space is valid. Notably, each feature's constraint $\epsilon_i$ on the local space varies; larger feature values usually imply greater activation but also indicate lower sensitivity to changes, warranting a larger local space. We use the mapping $\epsilon_i = \frac{x_i}{s}$ to make the constraints linearly related to the size of features. Our experiments will analyze the difference between using constant $\epsilon_i$ and linearly related $\epsilon_i$.

Next, we introduce our Local Attribution algorithm and present our core theorem:

**THEOREM 3.2 (LA).** *Given parameters $w \in \mathbb{R}^n$ and corresponding sample $x$, the local attribution for dimension $i$ can be expressed as*

$$LA(x_i) = \mathop{\mathbb{E}}_{\hat{x}=u(\tilde{x}),\tilde{x}\sim B_{\frac{\epsilon}{2}}(x)} \left[ (\hat{x}_i - \tilde{x}_i) \cdot \frac{\partial L(\tilde{x}_i; y, w)}{\partial \tilde{x}_i} \right] \tag{2}$$

*where $B_{\frac{\epsilon}{2}}(x)$ denotes the $\frac{\epsilon}{2}$-Local Space of sample $x$, and $u$ represents the exploration function.*

We define the importance of each dimension in the sample point based on the expected change in the loss function value caused within the local space. Intuitively, the dimensions that cause greater changes in the loss function within the effective space (local space) are more sensitive. Next, we will present the derivation proof from the expected change in the loss function to Eq. 2.

PROOF. Consider the expansion of the loss function $L$:

$$L(\tilde{x}; y, w) = L(x; y, w) + (\tilde{x} - x) \cdot \frac{\partial L(x; y, w)}{\partial x} + O \tag{3}$$

where $O$ represents higher order infinitesimals. From the property of expectation and the symmetry of the local space, we have:

$$\mathop{\mathbb{E}}_{\tilde{x}\sim B_{\frac{\epsilon}{2}}(x)} \left[ L(\tilde{x}; y, w) - L(x; y, w) \right]$$

$$= \mathop{\mathbb{E}}_{\tilde{x}\sim B_{\frac{\epsilon}{2}}(x)} \left[ (\tilde{x} - x) \frac{\partial L(x; y, w)}{\partial x} \right] \tag{4}$$

$$= \mathop{\mathbb{E}}_{\tilde{x}\sim B_{\frac{\epsilon}{2}}(x)} \left[ (\tilde{x} - x) \right] \cdot \frac{\partial L(x; y, w)}{\partial x} = 0$$

because $\frac{\partial L(x;y,w)}{\partial x}$ is independent of the choice of $B_{\frac{\epsilon}{2}}(x)$.

Firstly, we perform a first-order Taylor expansion of the loss function calculated for sample $x$ to get Eq. 3, which is substituted into Eq. 4 to calculate the expected transformation of the loss function within the local space. (Taking one dimension of $x$, replacing $x$ with $x_i$ in the formula, the derivation process remains unchanged, i.e., $\mathop{\mathbb{E}}_{\tilde{x}\sim B_{\frac{\epsilon}{2}}(x)} [L(\tilde{x}_i; y, w) - L(x_i; y, w)] = 0$). We see that under a first-order approximation, it is not possible to evaluate each feature through a single local space sampling. Introducing higher-order approximations can mitigate this issue, but due to the presence of sampling, it is impractical to introduce finite differences [15] to approximate the Hessian matrix in intermediate computations, which also makes introducing higher-order approximations computationally infeasible.

$$\mathop{\mathbb{E}}_{\hat{x}=u(\tilde{x}),\tilde{x}\sim B_{\frac{\epsilon}{2}}(x)} \left[ L(\hat{x}; y, w) - L(x; y, w) \right]$$

$$= \mathop{\mathbb{E}}_{\hat{x}=u(\tilde{x}),\tilde{x}\sim B_{\frac{\epsilon}{2}}(x)} \left[ (\tilde{x} - x) \frac{\partial L(x; y, w)}{\partial x} + (\hat{x} - \tilde{x}) \frac{\partial L(\tilde{x}; y, w)}{\partial \tilde{x}} \right]$$

$$= \mathop{\mathbb{E}}_{\hat{x}=u(\tilde{x}),\tilde{x}\sim B_{\frac{\epsilon}{2}}(x)} \left[ (\hat{x} - \tilde{x}) \frac{\partial L(\tilde{x}; y, w)}{\partial \tilde{x}} \right] \tag{5}$$

$$= \sum_{i=1}^{n} \mathop{\mathbb{E}}_{\hat{x}=u(\tilde{x}),\tilde{x}\sim B_{\frac{\epsilon}{2}}(x)} \left[ (\hat{x}_i - \tilde{x}_i) \cdot \frac{\partial L(\tilde{x}; y, w)}{\partial \tilde{x}_i} \right]$$

□

To enable practical computation, we introduce an exploration function $u$, which ensures that the transformed samples remain within the $\epsilon$-Local Space (proof in Appendix C). Inspired by MFABA [35] and AGI [17], the function $u$ can utilize both untargeted (Eq. 6) and targeted (Eq. 7) adversarial attacks, where the choice of $y^t$ is from categories other than the most probable, selected in descending order of confidence. Our experiments will involve an ablation study on the number of categories selected.

$$u^u(\tilde{x}) = \tilde{x} + \frac{\varepsilon}{2} \cdot \text{sign}\left( \frac{\partial L(\tilde{x}; y, w)}{\partial \tilde{x}} \right) \tag{6}$$

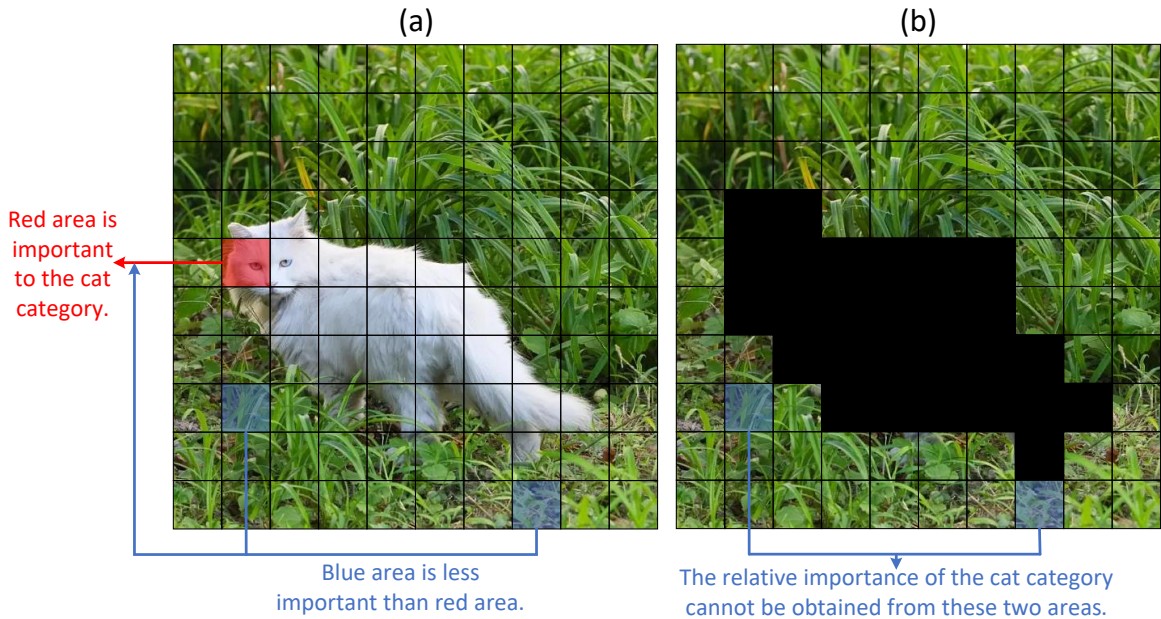

**Figure 3: After removing the important features, the relative importance of the remaining features is not as significant. As shown in (a), the features in the red area are notably more important for the category of cats compared to those in the blue area. However, as depicted in (b), after the cat features have been removed, it becomes challenging to assess the importance of the remaining features.**

$$u^t(\tilde{x}) = \tilde{x} - \frac{\epsilon}{2} \cdot \text{sign}\left(\frac{\partial L(\tilde{x}; y^t, w)}{\partial \tilde{x}}\right) \qquad (7)$$

### 3.3 Deep Analysis of Untargeted and Targeted Adversarial Attacks

The direct output of our neural network is defined as $z = f(x) \in \mathbb{R}^c$, and after passing through a softmax function, $z$ becomes a probability distribution $p = \text{softmax}(z) \in \mathbb{R}^c$ with $p_i \in (0, 1)$.

Observing the gradient $\frac{\partial z_i}{\partial x} \in \mathbb{R}^n$, updating $x$ along the direction of $\frac{\partial z_i}{\partial x}$ increases $z_i$ (proof refers to Eq. 3, the Taylor expansion). We examine the gradient information of $z$ during the computation of cross-entropy loss as shown in Eq. 8.

$$\frac{\partial L(x; y, w)}{\partial z_i} = \begin{cases} p_i - 1 & \text{if } i = \text{class of } y \\ p_i & \text{otherwise} \end{cases} \qquad (8)$$

Using the chain rule for gradients, we find $\frac{\partial L(x;y,w)}{\partial x} = \frac{\partial L(x;y,w)}{\partial z_i} \cdot \frac{\partial z_i}{\partial x}$. Combining with Eq. 8, we observe that when $i$ is the original category (the most probable category), the gradient information $\frac{\partial z_j}{\partial x}, j \neq i$, will be low, since the probability values $p_j$ are lower than for the original category. Thus, relying solely on untargeted adversarial attacks to explore the local space might neglect the information from categories other than the original. This necessitates the introduction of targeted attacks for other categories. As shown in Figure. 1, considering that the sign of targeted adversarial attacks is opposite to that of untargeted attacks, analyzing with $1 - p_j$

becomes relevant, where $1 - p_j$ is large when $p_j$ is small, allowing the preservation of gradient information $\frac{\partial z_j}{\partial x}$. From a gradient perspective, it is crucial to incorporate both forms of adversarial attack in the local space, and since $\epsilon_i$ remains the same under both attack conditions, their effects can be combined additively.

### 3.4 Local space sampling optimization

Finally, for the sampling process from $B(x)$ to obtain $\tilde{x}$, we can approximate it iteratively, using the gradient calculated from the previous sample step to perform a one-step attack from the original sample. If the sign of the gradient in the same dimension changes within a local space, it indicates that the dimension is sensitive and requires further exploration. If the dimension remains unchanged, it implies that maintaining the current dimension does not require alteration, thus reducing the scope of space that random sampling needs to explore. The obtained $\tilde{x}$ still resides within the local space $B(x)$, and we provide rigorous proof in Appendix D and pseudocode in Appendix E.

## 4 EXPERIMENTS

In this section, we provide a detailed description of the series of experiments conducted using the Local Attribution (LA) algorithm, including the choice of datasets, models, baseline methods, evaluation metrics, and experimental analysis.

**Table 1: Performance comparison of LA with 11 other competing methods across four models using Insertion and Deletion Scores. Higher Insertion and lower Deletion indicate better attribution performance, with Insertion considered more significant than Deletion**

| Method | Inception-v3 | | ResNet-50 | | VGG16 | | MaxViT-T | |
|---|---|---|---|---|---|---|---|---|
| | Insertion | Deletion | Insertion | Deletion | Insertion | Deletion | Insertion | Deletion |
| FIG | 0.05604 | 0.08542 | 0.03165 | 0.04278 | 0.02495 | 0.03880 | 0.23969 | 0.28277 |
| DeepLIFT | 0.09273 | 0.06974 | 0.04469 | 0.03378 | 0.03969 | 0.02343 | 0.26163 | 0.26138 |
| GIG | 0.10591 | 0.03879 | 0.05059 | 0.02005 | 0.04236 | 0.01649 | 0.29247 | 0.19346 |
| IG | 0.10863 | 0.04546 | 0.05802 | 0.02837 | 0.04461 | 0.02166 | 0.32399 | 0.26316 |
| SG | 0.18743 | 0.03688 | 0.12434 | 0.02316 | 0.12690 | 0.01746 | 0.46441 | 0.16277 |
| BIG | 0.20548 | 0.09443 | 0.12242 | 0.07208 | 0.08349 | 0.05596 | 0.36900 | 0.26257 |
| SM | 0.31201 | 0.10237 | 0.13429 | 0.08475 | 0.09684 | 0.06357 | 0.30256 | 0.24372 |
| MFABA | 0.32255 | 0.09913 | 0.14623 | 0.08333 | 0.11410 | 0.06083 | 0.28051 | 0.42919 |
| EG | 0.34311 | 0.28816 | 0.27563 | 0.22065 | 0.27820 | 0.35596 | 0.49227 | 0.55472 |
| AGI | 0.40435 | 0.08678 | 0.41482 | 0.06224 | 0.32855 | 0.05438 | 0.52116 | 0.24486 |
| AttEXplore | 0.44321 | 0.08062 | 0.32366 | 0.05471 | 0.29926 | 0.04445 | 0.40683 | 0.25082 |
| LA (ours) | 0.54415 | 0.07366 | 0.51956 | 0.04856 | 0.42071 | 0.04318 | 0.67147 | 0.23326 |

## 4.1 Dataset and Models

Our experiments randomly selected 1000 images from the ImageNet dataset, following the precedent set by existing methods such as AGI [17], MFABA [35], and AttEXplore [34]. Furthermore, we tested the LA algorithm using four different convolutional neural network architectures to assess its effectiveness and generality, namely Inception-v3 [26], ResNet-50 [7], VGG16 [23], and MaxViT-T [27].

## 4.2 Baselines

To comprehensively evaluate the performance of the Local Attribution (LA) algorithm and ensure the fairness of our assessments, we have compared LA against eleven baseline methods. These baselines cover a wide range of XAI methods, including AGI [17], AttEXplore [34], BIG [31], DeepLIFT [21], EG [5], FIG [8], GIG [10], IG [25], MFABA [35], SG [24], and SM [22]. These methods represent various technical approaches in the field of model explainability, providing a broad reference standard for evaluation.

## 4.3 Evaluated Metrics

We employed two traditional metrics, Insertion Score and Deletion Score, to assess the explanatory power of each explainability method. These metrics evaluate the model's dependence on different parts of the input data by analyzing changes in model performance [17]. The Insertion Score evaluates changes in model performance by progressively converting pixels from a baseline state (usually a state containing no meaningful information, such as all-black or all-white images) to the pixels of the original image. This conversion involves incorporating a certain number of the most important pixels, as determined by the explainability method, from the baseline state to their values in the original image. This process is repeated, re-evaluating the model's performance each time until all pixels have been converted from the baseline state

to their corresponding pixels in the original image. The Insertion Score is typically quantified by the degree of improvement in model performance, represented by the following formula:

$$\text{InsertionScore} = \frac{1}{N} \sum_{i=1}^{N} (P_i - P_0)$$

where $N$ is the total number of steps, $P_i$ is the model performance score after step $i$, and $P_0$ is the initial state performance score (when no significant pixels are inserted).

The Deletion Score is calculated by progressively removing the most important pixels from the original image and observing changes in model performance. Each removal involves replacing a certain number of the most important pixels (again determined by the explainability map) with pixels of a baseline state. This process is continuously repeated, re-evaluating the model's performance each time, until all pixels deemed important have been removed. The Deletion Score is quantified by the degree of reduction in model performance, represented by the following formula:

$$\text{DeletionScore} = \frac{1}{N} \sum_{i=1}^{N} (P_{\text{full}} - P_i)$$

where $P_{\text{full}}$ is the model performance score under the full image, and $P_i$ is the model performance score after step $i$ when important pixels have been removed.

We identified implementation biases in the Insertion Score and Deletion Score in the open-source codes of RISE [18], MFABA [35], BIG [31], and AGI [17]. Previous works performed importance ranking by sorting each channel of the image separately, but the actual evaluation process should treat each input dimension equivalently. Therefore, we corrected this bias in this paper. To ensure consistency in experimental results, we also replicated the experiments

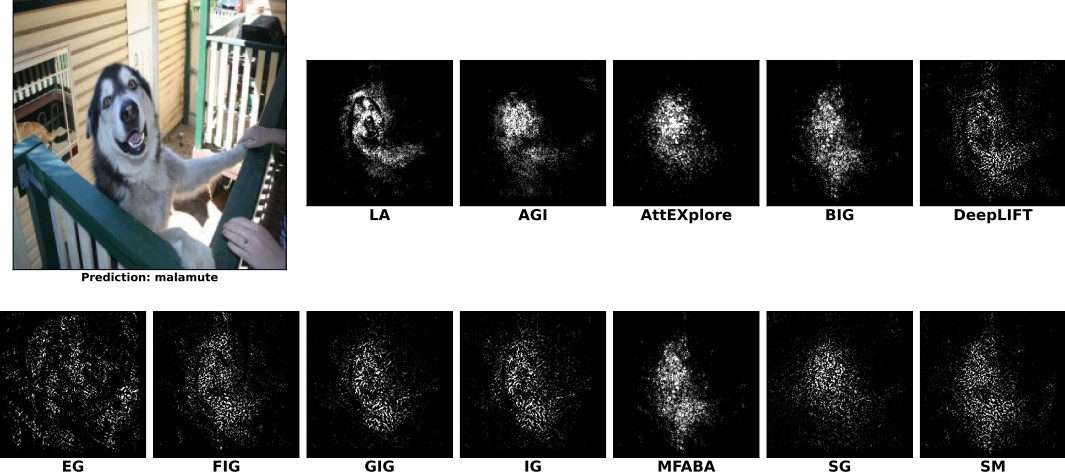

**Figure 4: Visual comparison of the attribution effects of LA and other competing algorithms on the Inception-v3**

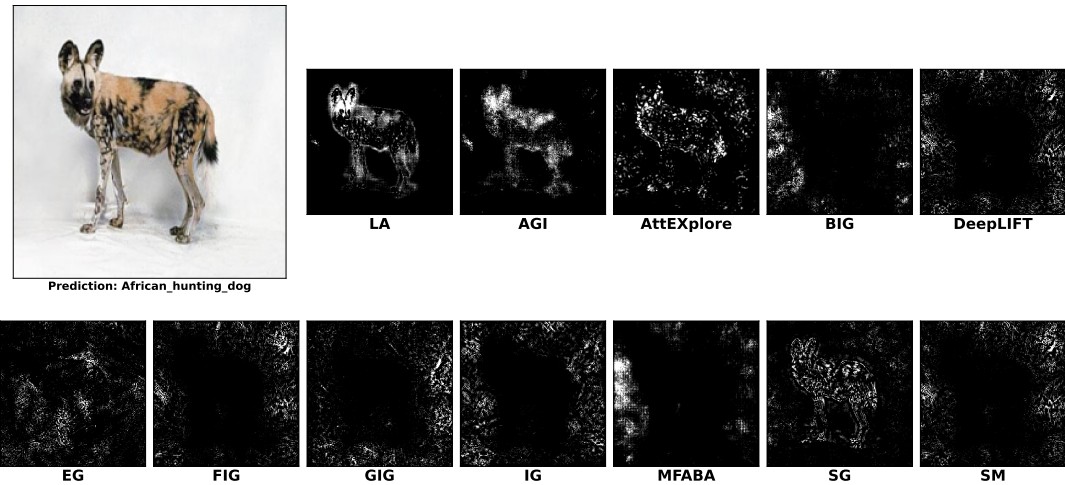

**Figure 5: Visual comparison of the attribution effects of LA and other competing algorithms on the MaxViT-T**

from previous works (presented in Appendix F). Notably, prioritizing the insertion of important features with the Insert score and prioritizing the removal of unimportant features with the Deletion Score are equivalent, as known by symmetry. Furthermore, according to **RQ2**, when important features are lost, the attribution results are less meaningful, implying that a Deletion score focused on removing important features only needs to be within a minimal range and is less informative than prioritizing the addition of important features with the Insertion score.

## 4.4 Parameters

In this series of experiments, we set the number of sampling to 30 for the MaxViT-T model and 20 for the other models. The spatial range $s$ was consistently set to 20 across all experiments.

## 4.5 Experimental Results

As shown in Tab. 1, our LA method achieved significant improvements. Compared to other methods, the average increase in Insertion across the four models was 0.31758, and the average reduction in Deletion was 0.028883. Specifically, the average improvements in Insertion for the Inception-v3, ResNet-50, VGG16, and MaxViT-T models were 0.30948, 0.36262, 0.28626, and 0.31197 respectively; while the reductions in Deletion were 0.019774, 0.017429, 0.025277, and 0.053052 respectively. Compared to the latest attribution methods like AGI, MFABA, and AttEXplore, LA showed clear advancements. Notably, LA not only improved in terms of Insertion but also reduced Deletion, indicating a comprehensive enhancement in explainability performance compared to these methods. While some methods slightly outperformed LA in terms of Deletion, their Insertion scores were substantially lower than LA. As previously mentioned, the significance of Insertion outweighs that of Deletion,

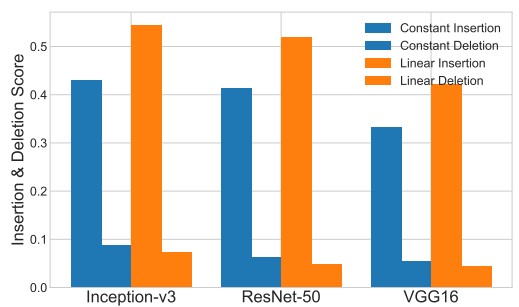

**Figure 6: Comparison of Insertion and Deletion Scores across Different Models and Space Constraints**

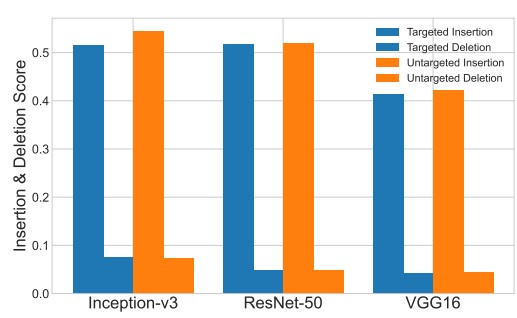

**Figure 7: Comparison of Insertion and Deletion Scores across Different Models and Attack Type**

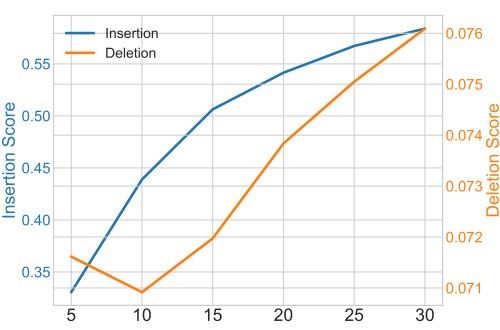

**Figure 8: The impact of changes in sampling times on the performance of the LA method on the Inception-v3**

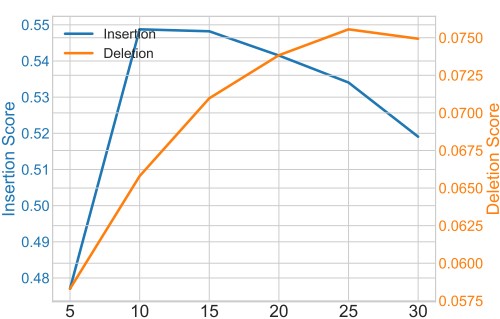

**Figure 9: The impact of changes in Spatial Range $s$ on the performance of the LA method on the Inception-v3 model**

thereby firmly establishing the efficacy of the LA method. More results are provided in Appendix F and G.

Additionally, Figure. 4 and Figure. 5 show the attribution results of our LA method versus other methods on the Inception-v3 and MaxViT-T models. It is evident that our LA method more accurately and concisely captures the key features in images, while the outputs from other methods appear more dispersed and blurred.

## 4.6 Ablation Studies

*4.6.1 Impact of Constant vs. Linear Space Constraints on Effectiveness.* This section discusses the impact of Constant and Linear space constraints on the effectiveness of LA. We fixed the number of samples and the spatial range $s$ at 20. As shown in Figure. 6, across different models, the attribution performance under Linear space constraints was comprehensively better than under Constant constraints, with significantly higher Insertion Scores and lower Deletion Scores under Linear constraints.

*4.6.2 Impact of Attack Type on Effectiveness.* We discuss the impact of targeted and untargeted attacks on the effectiveness of LA. The number of samples and the spatial range $s$ were kept constant at 20. As depicted in Figure. 7, across different models, the Insertion from Untargeted Attacks was higher than from Targeted Attacks. However, the Deletion Scores were relatively similar, showing little variation.

*4.6.3 Impact of Sampling Times on Effectiveness.* This part discusses how the number of Sampling affects the effectiveness of LA.

We kept the spatial range $s$ at 20. As shown in Figure. 8 with an increase in sampling rate, LA's Insertion Score increased and showed a trend towards convergence, while the increase in Deletion Score was more abrupt.

*4.6.4 Impact of Spatial Range $s$ on Effectiveness.* In this section, we explore the impact of the spatial range $s$ on the effectiveness of LA. The number of samples was fixed at 20. As illustrated in Figure. 9, with an increase in $s$, the Insertion Score initially increased and then decreased, peaking when $s$ was at 10. Conversely, the Deletion Score increased with larger $s$ values, but the rate of increase gradually weakened.

## 5 CONCLUSION

In this paper, we identifies the challenge of ineffective intermediate states in current attribution algorithms, which has significantly impacted the attribution results. To better investigate this issue, we introduces the concept of Local Space to ensure the validity of intermediate states during the attribution process. With these findings, we propose the LA algorithm, which can comprehensively explore the Local Space using both targeted and untargeted adversarial attacks, thereby achieving state-of-the-art attribution performance in comparison with other methods. We provide rigorous mathematical derivations and ablation study to validate the significance of each component in our algorithm. We anticipate this work will facilitate the attribution method in XAI research.

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
