# OpenReview forum: "Enhancing Model Interpretability with Local Attribution over Global Exploration"
_acmmm.org/ACMMM/2024/Conference — MM2024 Poster_

### Official Review · Reviewer_6EBM · 2024-05-24

**Rating:** 4
**Confidence:** 3

**Summary:**

The focus on local spaces to avoid OOD issues is innovative and addresses a critical problem. The paper is well-executed, with strong experimental results and an interesting approach to improving attribution accuracy. While the paper provides mathematical derivations and proofs of the Local Attribution (LA) method’s compliance with attribution axioms, it could benefit from a more in-depth theoretical analysis. Specifically, exploring the limitations and potential failures of the method in different OOD scenarios would add significant value.

**Strengths:**

1.	The paper addresses the critical issue of OOD impacts in model interpretability, which is highly relevant in the field of AI. Understanding and mitigating OOD effects is essential for the robustness and reliability of AI models.

2.	The experimental results are impressive, demonstrating a significant improvement in attribution effectiveness compared to state-of-the-art methods. The reported 38.21\% average improvement is notable and suggests that the proposed LA algorithm is highly effective.

3.	The idea of focusing on local spaces for attribution rather than global exploration is intriguing. This approach helps to avoid the pitfalls of OOD spaces, potentially leading to more accurate and meaningful attributions.

**Limitations:**

1.	The theoretical aspects of the paper are somewhat trivial. While the concept of local spaces and their properties is well-presented, the paper does not provide a deep theoretical analysis of the method's generalization capabilities in OOD scenarios. A more rigorous theoretical exploration could strengthen the paper’s contribution.

2.	The paper does not sufficiently analyze how the proposed Local Attribution method generalizes to various OOD problems. A thorough discussion on the method's limitations and potential generalization issues in different OOD contexts would provide a more comprehensive understanding of its applicability and robustness.

**Suitability:**

2

---

### Official Review · Reviewer_urs9 · 2024-05-24

**Rating:** 4
**Confidence:** 3

**Summary:**

This paper argues that existing attribution algorithms suffer from a significant drawback of generating numerous intermediate states that reside in the out-of-distribution (OOD) space, leading to compromised attribution results. To address this issue, the authors propose a Local Attribution (LA) method, which leverages targeted and untargeted adversarial attacks to explore the local space. The paper provides comprehensive theoretical proofs and derivations for the proposed method and empirically demonstrates its state-of-the-art performance.

**Strengths:**

1. The paper effectively highlights the limitations of current attribution algorithms and presents theoretical evidence supporting the efficacy of the proposed Local Attribution (LA) method.

2. The writing is coherent, and the logical flow is well-maintained. Furthermore, the experimental validation substantiates the effectiveness of the proposed approach.

3. The idea introduced in the paper exhibits novelty and is logically sound.

**Limitations:**

1. The authors utilize Figure 3 to illustrate that the assessment of feature importance becomes meaningless when no significant features are present. However, it would be beneficial to provide theoretical justification for this claim, as it may relate to the distribution of image data.

2. In the ablation experiments, the authors investigate the impact of different conditions, such as Space Constraints and Attack Type, on the effectiveness of the LA method. However, further explanations or insights into the experimental results are lacking.

3. The figures included in the paper appear somewhat simplistic, lacking the expected academic quality. Enhancing the visual representations would enhance the overall presentation. Also, the state-of-the-art results achieved by the LA method might to be emphasized in Table 1 (e.g., by bolding the relevant values).

**Suitability:**

2

---

### Official Review · Reviewer_nBeQ · 2024-05-24

**Rating:** 5
**Confidence:** 2

**Summary:**

The proposed Local Attribution (LA) algorithm aims to provide precise explanations by focusing on local space properties rather than exploring the entire sample space, which can lead to intermediate states in the out-of-distribution (OOD) space. The LA algorithm incorporates both targeted and untargeted adversarial attacks to explore local spaces effectively. Experimental results demonstrate a significant improvement in attribution effectiveness compared to state-of-the-art methods.

**Strengths:**

1. The concept of local space for attribution and the dual-phase exploration approach of LA is a novel contribution that enhances the precision and reliability of model interpretability.
2. The use of targeted and untargeted adversarial attacks to comprehensively explore local spaces is technically sound and innovative. The detailed mathematical derivations and proofs of compliance with attribution axioms strengthen the methodological foundation.
3. The paper provides extensive experiments comparing LA with various state-of-the-art methods across multiple models and metrics.
4. The paper is well-structured, clearly explaining the proposed method.

**Limitations:**

The proposed method involves computationally intensive adversarial attacks and multiple phases, which increases the training overhead and complexity. This aspect is not discussed in detail in the paper.

**Suitability:**

2

---

### Official Review · Reviewer_VART · 2024-05-25

**Rating:** 4
**Confidence:** 3

**Summary:**

This paper investigates the problem of attributing a model's output w.r.t. its input features, one of the core problems in explainable AI. The paper identifies an important issue overlooked by previous studies in this area, namely, the intermediate states used in attribution may be out-of-distribution (OOD) and thus lead to unreliable attribution results. The paper then proposes to search for intermediate states in a _local_ region of inputs (i.e., local space). The notion of local space is proposed by this work, while the search function (i.e., exploration function) is instantiated to be targeted or untargeted adversarial attacks, which is also used in prior work such as MFABA and AGI.

**Strengths:**

- The problem identified by the paper (OOD intermediate states) is intuitive and well-motivated.

- The proposed notion of local space and the corresponding algorithm is simple and seems easy to implement.

- Experimental results are generally good with sufficient ablations.

**Limitations:**

- I think the proof in Appendix A about the _sensitivity_ axiom is incorrect: according to the original definition of sensitivity in IG [1], sensitivity refers to "if for every input and baseline that differ in one feature but have different predictions then the differing feature should be given a non-zero attribution" [1, Section2.1]. The authors of [1] also give a counterexample suggesting why vanilla gradients violate this axiom: the neural network may be "flat" around the input (but not flat around the baseline), leading to zero gradients at the input and hence zero attribution. Since your method also searches over a **local** region of the input, it may also encounter such a "flatness" issue which leads to zero attribution. E.g., when $\epsilon\to 0$, the expected gradient $\mathbb{E}(\frac{\partial L(\tilde{x}; y, w)}{\partial \tilde{x}})$ for $\tilde{x}\sim B_{\epsilon / 2}(x)$ would be essentially the same as the vanilla gradient $\frac{\partial L(x; y, w)}{\partial x}$. So in this scenario, your method would also lead to zero attribution if $\frac{\partial L(x; y, w)}{\partial x}\approx 0$.

- Besides the selectivity axiom, [1] also introduces another axiom of _completeness_, which is not covered in the proof. It would be great if the authors could discuss whether your method satisfies this axiom or whether satisfying this axiom is necessary.

- The authors state that "An intermediate state is in the OOD space if it cannot maintain the same model decision as the original state." (L375-377), which does make sense to me. However, such a criterion is not used in the definition of the local space nor explored in the ablation; instead, the authors resort to manually tuning the "size" of the local region by introducing a hyperparameter. Why not directly introduce the model decision as a criterion in searching for non-OOD intermediate states?

- To be honest, I didn't really get the relation between the RQ2 in the paper and the notion of local space in your method. The paragraph starting from Line 368 to Line 379 seems quite confusing in connecting those two concepts.

- The writing of the paper can be improved. E.g., I think the first several paragraphs in the introduction are kind of lengthy and contain some contents that are not directly related to the paper (e.g., the application of AI in various fields). Also the notion of "intermediate states" can be introduced in a way that is easier to understand, e.g., you can include the formula of IG [1] in Section 3.2 to show how intermediate states contribute to attribution before giving the example.


[1] Axiomatic Attribution for Deep Networks.

**Suitability:**

2

---

### Meta-Review · Area_Chair_9FNB · 2024-06-29

**Recommendation:** Accept (Poster)
**Confidence:** 4

**Metareview:**

This paper contends that existing attribution algorithms suffer from a notable limitation: they generate numerous intermediate states that fall into the out-of-distribution (OOD) space, thereby compromising attribution accuracy. To tackle this challenge, the authors propose the Local Attribution (LA) method, which uses targeted and untargeted adversarial attacks to explore the local input space. The paper includes comprehensive theoretical proofs and derivations for the proposed method, and empirically demonstrates its state-of-the-art performance.

All reviewers found the paper interesting with clear motivation and presentation. The experimental results are sound. I would encourage the authors to incorporate the comments and suggestions in the final version.